# How Much Dialogic Coordination Practices Matter to Healthcare Professionals—A Delphi Approach towards a Tool for Identification and Measurement

**DOI:** 10.3390/healthcare11222961

**Published:** 2023-11-14

**Authors:** Mónica Santos-Cebrián, Miguel-Ángel Morales-Moya, Carmen De-Pablos-Heredero, María-del-Rosario Pacheco-Olivares

**Affiliations:** 1Financial Economy and Accounting Department, Rey Juan Carlos University, 28032 Madrid, Spain; monica.santos@urjc.es (M.S.-C.); ,; 2Department of Business Economics (Administration, Management, and Organization), Applied Economics II and Fundamentals of Economic Analysis, Rey Juan Carlos University, 28032 Madrid, Spain; carmendepablos@urjc.es

**Keywords:** dialogic coordination practices (DP), relational coordination (RC), health teamwork, content validation, health policy, fast-response organisations, communication networks, Delphi method

## Abstract

The study of coordination practices in health policy is a central aspect. The need for further research has been recently highlighted because of COVID-19. In this sense, dialogic practices (DP) have been identified but not validated yet. The purpose of this study is to develop and validate a DP questionnaire for healthcare teams. Items were identified based on a literature review, and the content validation was carried out by means of a Delphi study. A total of 10 experts assessed the clarity and appropriateness of the items and their corresponding measurement scales. After two rounds, a high level of consensus was reached, with agreement of 90% or higher on all items, and a high degree of stability and concordance in the results. This study resulted in a questionnaire consisting of four items, one for each identified DP initially proposed to the experts, as no other practices were revealed. From a practical perspective, the validation of these items constitutes a methodological innovation that responds to the call in the literature to open new avenues for comparative studies, and the possibility of generalising the findings and bringing together different approaches to the problem of coordination, which is key in health policy where unforeseen situations emerge.

## 1. Introduction

Work coordination has been addressed from different disciplines and perspectives in the literature and is considered a central aspect of organisational management and crisis and disaster management [1,2]. The more traditional approaches which have emerged in organisational theory assumed that the environment is predictable and that coordination should take place through stable, pre-designed coordination structures and mechanisms. Changes in the nature of work, the proliferation of fast-response organisations (FROs) and the extraordinary importance of coordination in extreme and emergency situations prompted researchers to adopt a new approach based on on-site observations of coordination practices in different contexts and organisations [3,4]. This approach views coordination as an emergent, contextual, and situated phenomenon in which coordination mechanisms are the result of “dynamic social practices” that are continuously constructed and reconstructed, rather than stable entities [5] (p. 907). One of the definitions of coordination which best represents this approach and is most widely accepted among researchers is provided by Faraj and Xiao [6] (p. 1157) as the “temporally unfolding and contextualised process of input regulation and interaction articulation to realise a collective performance”.

However, the obstacles identified by Okhuysen and Bechky [4] to developing a unified theoretical framework on coordination still persist: (a) different approaches to the problem of coordination depending on the disciplinary field from which the research is approached; (b) the difficulty in comparing and generalising empirical findings obtained in different case studies due to their rootedness in particular contexts and the great terminological diversity used by researchers to describe similar mechanisms, practices, and processes; and (c) a lack of explanation of how coordination occurs in real time.

Numerous studies are making important advances on how coordination occurs in groups and organisations facing legislative and structural changes [5], frequent surprises [7], uncertainty [8], new tasks and problem-solving [9,10], extreme situations, emergency and disaster management [1,2,11,12,13,14,15,16,17,18,19], and FRO coordination [6,20]. However, no progress has yet been made in overcoming the remaining obstacles. The great diversity of terminology, the different approaches to the problem of coordination acting autonomously, and the virtual absence of comparative studies and generalisable findings make the goal of obtaining a unified theoretical framework a long way off. 

For example, although in healthcare settings the academic concept of dialogic practice may not be widely recognized, we often encounter situations where there is no specific protocol to guide action or where the existing protocol is not effectively applied. In these cases, healthcare professionals must turn to dialogic practices and relational coordination to make informed decisions and provide the best possible care to patients. This may involve open discussions about treatment options, risk and benefit assessment, and shared decision making. In unprecedented situations, relational coordination becomes an essential resource, as it enables medical teams to adapt swiftly and effectively to changing circumstances. The presence of multiple approaches and theories on coordination in the healthcare sector, from traditional hierarchical models to more collaborative approaches based on the involvement of multiple stakeholders, makes it challenging to identify specific dialogic coordination practices that fit each context.

The COVID-19 pandemic has highlighted these shortcomings. According to [21], although coordination is recognised as a central aspect in the literature on health system resilience and emergency management, there is very little scientific evidence on the subject. At the intra-organisational and team level, this also occurs [22,23], but there is clear evidence of the critical role of dialogue in the process of knowledge transformation and integration, the important role of informal coordination practices in contexts of high uncertainty, and the alternation between formal and informal coordination practices in FRO [5,6,8,10,24]. 

Although the pandemic has come as a shock to all levels and sectors of activity, the health sector has been the most affected due to the huge increase in demand for its services, the scarcity of resources and scientific evidence, and the threat to the health of workers themselves. This scenario is a hyperbole of the extreme situations in which the problem of coordination in healthcare organisations has been studied. However, studies on the response capacity of hospitals to the pandemic are anecdotal [23]. 

Two important practice-based approaches to the coordination problem are relational coordination (RC) and dialogical coordination practices (DP). 

RC is defined as “a mutually reinforcing process of communicating and relating for the purpose of task integration” [25] (p. 301). Today, RC is both a theory and a set of analytical methods aimed at understanding the relational dynamics of work coordination, both within and between organisations [26] (p. 16). This theory has been applied and empirically validated across different sectors of activity and geographical areas around the world, especially in the healthcare sector [27]. It argues that when coordination takes place through frequent, quality, and problem-oriented communication and is based on mutually respectful relationships with shared goals and knowledge, organisations are better able to achieve the desired results [26,28]. RC influences the quality of teamwork [29,30] and fosters organisational flexibility to adapt to changing, highly uncertain, and interdependent environments [31]. In this regard, recommendations for US health service managers to increase the resilience of healthcare facilities during and after the COVID-19 pandemic are summarised as improving RC [32] (p. 9).

DPs fall under the umbrella of informal coordination practices and were identified in the influential work of Faraj and Xiao in a trauma hospital [6]. The authors identified two types of practices: expertise practices and DPs. Unlike the former, DPs occur in unforeseen, complex, and urgent situations that require a rapid response and in which the usual procedures (manuals, protocols, and routines) are not sufficient or adequate to resolve the situation. DPs are reactions and decisions made on the fly in response to the evolution of problematic, unforeseen, and complex situations which challenge the mental models of the work team and whose resolution requires dialogue between different specialities and/or professional groups. Four types of DPs were identified: epistemic contestation, joint sensemaking, cross-boundary intervention, and protocol breaking. The management of the COVID-19 emergency led to an intensification of DP deployment [33] (p. 166). 

This paper is aimed to show preliminary results of a research project oriented to analyse the impact of the COVID-19 pandemic on healthcare teamwork coordination in a hospital in Spain. 

One of the proposals of this research project is that RC and DPs are related and mutually reinforcing. Although at the theoretical level, several authors have suggested links between RC and DPs (i.e., refs. [34,35] suggested that RC is an inducer of DPs in organisations), to the best of our knowledge, this relation has not yet been empirically explored in the literature. 

Furthermore, ref. [10] shows how DPs are used for “transcending” knowledge gaps in novel and urgent situations between different communities of practice and how these practices condition the way they communicate and relate to each other in the future. However, the relationship between RC and DPs needs to be further investigated at both theoretical and empirical levels. To this end, it is necessary to develop a set of DP metrics which are compatible with existing RC metrics in order to propose and validate an improved RC model that takes into account the influence of the DPs. 

Based on the above, this work aims to develop and validate a set of items on the four types of DPs, previously identified in the literature, through which it will be possible to identify the use (deployment) and frequency of such practices in hospital work teams in complex and unforeseen situations where the trajectory of events is not as expected and which require a rapid response.

This work has several implications. On the one hand, it will help healthcare FROs and their different professional groups to highlight the importance of dialogue and informal coordination practices in contexts of high uncertainty. On the other hand, it provides researchers in the field with a new tool that opens the door to comparative studies, the generalisation of findings obtained in case studies, and the possibility of establishing relationships between different ways of approaching the problem of coordination.

Through the validation of DP items and their integration with RC items already validated in the literature review, it will be possible to find out the linkages between RC and DP and their impact on final healthcare performance. This way, it will be easier for healthcare organisations to promote those coordination practices exhibiting the best results regarding teamwork quality, which will be translated into better healthcare outcomes. This paper is a first step to doing so. 

The remainder of this paper is structured as follows: Section 2 describes the Delphi methodology and the phases followed to validate the questionnaire developed around the DPs. Section 3 details the results obtained in the different rounds of expert consultation and their corresponding discussion. Finally, the most relevant conclusions reached in this study are presented.

## 2. Materials and Methods

The Delphi methodology is a widely used method in the context of research, especially in the field of health and social sciences [36,37,38]. Its usefulness for the validation of questionnaires has been demonstrated in many studies [39,40,41]. Through Delphi, valuable information on the clarity and relevance of the items included in the questionnaire can be obtained from the opinion of several experts on the topic being evaluated [41,42,43]. The Delphi technique is an iterative, controlled, and anonymous expert consultation process with statistical feedback on the results obtained in successive rounds [37].

The process of preparing and conducting a Delphi study involves several steps that are usually grouped into different stages. The number and naming of these stages vary in the literature. However, the step sequence is very similar, despite being grouped in different ways. In this paper, we have grouped the steps followed in the study into three phases, following [42,44]: preliminary, exploratory, and final.

Preliminary phase: the following steps were undertaken during this phase.

-Configuration of the coordinating group: the coordinating group is made up of the 4 authors of this study.-Literature review: an exhaustive search for papers related to the DPs identified by Faraj and Xiao [6] was carried out through direct observation and interviews. Although these practices have been widely cited in the literature on coordination, especially in FRO and emergency management, refs. [1,23,33] among others, there are hardly any replications of the empirical study conducted by these authors, nor have metrics been developed to measure the presence of DPs in the healthcare context. The review of the literature made it possible to recognise the particularities of dialogic coordination in the healthcare setting and, in this way, to formulate a group of items adapted to this scenario in relation to the DPs.-Development and review of the DP questionnaire to be validated; The coordinating team prepared the questionnaire, and after its revision, 8 items were included, 2 for each DP. These items were submitted for evaluation through the Delphi during the second half of July 2022.-Preparation of the questionnaire for the first round of the Delphi: in this questionnaire, the experts were asked for their opinion on the clarity and appropriateness of the items related to each DP and their corresponding measurement scales.-Selection of the panel of experts: The selection of experts was one of the key aspects for the validity of the Delphi results. In this sense, the criteria for selecting the experts and the number of experts selected depended on the subject matter to be addressed and the objective to be achieved in the application of the Delphi method [37,45,46]. In this case, the problem to be addressed and the scope of application were very specific. Two categories of experts from the health and academic fields were defined. In the healthcare field, experts were selected according to these criteria: (1) they are representative of the target population for the final questionnaire and (2) they are healthcare personnel with extensive professional experience working on the front line during the COVID-19 pandemic. In the academic field, experts whose research activity is related in one way or another to coordination and teamwork and/or hospital care were selected.

Considering that Delphi method-based studies do not require sample representativeness for statistical purposes, sample size is typically determined based on the nature of the research, the complexity of the problem, the homogeneity or heterogeneity of the sample, and the availability of resources. Health-related Delphi studies often have a relatively small sample size and recruit participants with specific knowledge of a condition. Given the specificity of our research and the homogeneity of the sample, a small panel of experts was chosen. 

The coordinating team invited 12 potential experts who met the selection criteria. In the end, 10 of them agreed to participate in this study. 

Table 1 shows the details of the experts who finally made up the panel.

-Pre-check of the questionnaire of the first round of the Delphi-DP: A consultation with two experts not included in the panel of experts, one with a health profile and the other with an academic profile, was conducted. In this pre-check, some aspects were detected that could be improved in relation to the measurement scales and terminology commonly used in the health sector. The suggestions were analysed and approved by the coordinating team.-Preparation of the final questionnaire for the first round of the Delphi: The above suggestions were incorporated into the questionnaire which was finally sent to the panel of experts. Before proceeding with the submission, the coordinating team made a video describing the purpose and schedule of the Delphi process. This video can be accessed in the Appendix A.

Exploratory phase: during this phase, two rounds of expert consultation were carried out to reach a consensus on the appropriateness and validity of the DP items and their measurement scale.

-First round: The questionnaire proposed by the coordinating team was sent out for the 10 experts to give their opinion on the appropriateness of the items chosen for the measurement of the DPs. The questionnaire was divided into four blocks corresponding to the four DPs. In each block, the expert was asked to indicate whether they believe that these questions correctly measure the aspects they are intended to measure. If they considered the question to be inadequate, they were asked to propose an alternative question and/or make any suggestions or appreciations they may have in this respect. At the end of the questionnaire, the expert was asked whether, based on their professional experience, they could identify other DPs not covered by this study. This first round was carried out during the week of 18–24 July 2022.-Second round: Once the responses had been processed and the overall results of the first round had been analysed, the coordinating team prepared a report with the results obtained in the first round. After analysing the comments and suggestions made by the experts, the questionnaire to be sent out in the second round was drafted, including information on the degree of agreement on each question and most of the suggestions that revolved around the terminology used. This second round was carried out during the week of 25–31 July 2022.

In this second round, experts were asked to reassess their responses in the light of new information obtained in the first round in the search for a consensus. 

The final phase: Once the responses had been processed and the overall results of the second round had been analysed, the coordinating team prepared a report with the results obtained in the second round. After analysing the comments and suggestions made by the experts, a consensus was reached on all the items and measurement scales, and the Delphi process was therefore concluded. 

As a result of the whole process, the definitive and validated DP items were generated and incorporated into the RC and DP questionnaire. In the second phase of the project, this questionnaire was launched to all healthcare staff in the hospital where the research project was carried out. 

Figure 1 shows an outline of the steps followed in the 3 phases which were carried out in this study.

We have included a variety of metrics to analyse the results of the Delphi method for both rounds, with the purpose of obtaining a comprehensive and robust insight into the generated information:For each of the items related to each DPS, the median (Mdn) the arithmetic mean (Mean), standard deviation (SD), coefficient of variation (CV), interquartile range (IQR), relative interquartile range (RIR), frequency of assessment (f), and l of consensus (CONS) have been calculate.To analyse the stability and concordance in the analysis of the results, a comparative analysis based on the variation of RIR and CV has been provided.

Each metric offers valuable information about central tendency, variability, consistency, and the level of consensus, collectively helping us better understand expert opinions and the quality of responses generated in this study.

Ethical aspects:

This study was approved by the Research Ethics Committee from the hospital that participated in this study. 

In accordance with the parameters established in Spanish Organic Law 3/2018, 5 December, on the Protection of Personal Data and the guarantee of digital rights, and the Declaration of Helsinki promulgated by the World Medical Association (WMA) in 1975, anonymity was preserved both in the application and feedback of the questionnaire and the acceptance of participation by experts both in the successive rounds of the Delphi and in the preliminary check in which two additional experts took part.

## 3. Results

Once the information for round one had been collected, the responses of the 10 experts on the eight items grouped in pairs around each of the four identified DPs were analysed: one on the appropriateness of the question about each DP and one on the appropriateness of the measurement scale. 

The experts responded on a Likert scale from 1 to 5, where 1 was strongly disagree, 2 disagree, 3 neither agree nor disagree, 4 agree, and 5 strongly agree.

Although there is no single way of determining when a consensus is reached among the different experts consulted in the Delphi, in this study, it is understood to be reached when the criteria established by the various authors cited in [46,47] are met. As shown in Table 2, for ordinal data, the median has been chosen as an indicator of central tendency, supported by the interquartile range (IQR) [46,48,49]. Furthermore, the median value is very close to the mean, indicating that the distribution is approximately symmetrical. 

Table 3 shows the results of the first round of the Delphi.

As can be seen in Table 3, for all questionnaire items in the first Delphi round, the median is equal to 4 (≥4 = agree), and this value remains in the second and third quartiles, indicating that most responses are concentrated in the Likert scale rating of 4 or 5 and again, confirming that the distribution is relatively symmetrical. The IQR reaches a maximum value of 1 (≤1.5). The frequency of values 4 and 5 (agree; strongly agree) has a minimum value of 80% (≥70%), reaching 100% in some cases. 

Therefore, the above data show that a consensus has already been reached among the experts in the first round of the Delphi, both on the appropriateness and clarity of the DP items, as well as on the measurement scales proposed. 

However, even though there was agreement, some experts made some suggestions to better adapt the questions to the language used by healthcare personnel. The coordinating group proceeded to carry out a new round that included the new wording of the questions, incorporating the suggestions of the experts set out in Table 4. 

As mentioned above, at the end of the questionnaire, the experts were asked whether, based on their professional experience, they could identify other DPs not covered by the study. One of the experts included a comment to this effect, that a “lack of communication between team members may prevent the exercise of dialogic practices in situations where it is necessary to call on the team to resolve them”. The expert recognises that there must be minimum levels of communication for the DPs to be deployed, in line with the proposal we made in the introduction to this paper and in agreement with [34,35].

In round II, only the questions regarding the DPs were included, as the questions regarding the scales of measurement reached full consensus in the first round. In this round, all experts who participated in the first round participated. The results of the second round are shown in Table 5. 

As shown in Table 5, the values of all of the statistical parameters improve with respect to the previous round, indicating that the redrafting of the questions following the suggestions of the experts has strengthened the degree of consensus among them. For items related to DP1, DP3, and DP4, the frequency of ratings 4 and 5 on the Likert scale is 100%, and for DP2, 90%. Applying the acceptance criteria, we observe that for all items, the median ≥4 and IQR ≤ 1.5 are met. 

In Table 6, the results of both rounds are compared in order to analyse the stability of the panel, which is conceived as the consistency in the experts’ opinions between successive rounds of the Delphi, regardless of their degree of convergence [50]. In this table, we compare the different parameters analysed in both rounds, showing how they all improve [6]. 

We understand that stability occurs if the variation of the interquartile range between rounds is less than 0.30, and a consensus is considered to be reached if the variation of the coefficient of variation between rounds is less than 0.40 [47], as shown in Table 7. 

Based on the results obtained, the Delphi is closed after the second round, given that the criteria for closing the Delphi are met, as there is a high degree of consensus (median and interquartile range) and great stability in the opinions of the experts between rounds (variation in RIR and CV between rounds).

## 4. Discussion and Conclusions

The main objective of this study was to develop and validate a questionnaire to identify the presence and frequency of the four types of DPs that occur in healthcare teams in complex and unforeseen situations when unexpected events occur and a rapid response is demanded: epistemic contestation, joint sensemaking, cross-boundary intervention, and protocol breaking. 

The validation of a questionnaire prior to its launch is essential to guarantee the quality, reliability, and validity of the data obtained, as well as to ensure that the questionnaire is appropriate and understandable for the population to be surveyed. The Delphi method is a sound research methodology, particularly useful in contexts where information and knowledge are limited or at a very exploratory context, especially in the social and health sciences. It has been widely used to validate the content of questionnaires of different natures. 

In this study, through two rounds of expert consultation following the Delphi method, the items of the DP questionnaire and their corresponding measurement scales have been improved and validated. A high degree of agreement, stability, and concordance has been reached from the statistical analysis of the results. The experts agreed with a consensus of over 80% for all items in the first round and 90% in the second round. Although in the first round, sufficient agreement was reached to validate the questionnaire, the experts were given the opportunity to include suggestions and comments in an open-ended manner. The second round allowed the questions to be reformulated in response to suggestions made by some of the experts to improve the appropriateness and clarity of the items included in the questionnaire on the DPs detected by [6] and adapted to the healthcare context, using terminology from the jargon used by healthcare professionals.

Although there are no standard criteria to assess the quality of the Delphi, the essential parameters to be applied during the expert consultation process have been met: selection and anonymity of experts, iteration, controlled feedback, and statistical stability of the consensus. 

As future research, deeper statistical analysis will be conducted when the full questionnaire incorporating the CR and PD questions is ready. Before launching the final questionnaire to the study population, a pre-test will be carried out with a sample of the population in which it will be subjected to complementary tests of reliability, validity, and internal consistency. 

From this research, it can be concluded that the validated questionnaire constitutes a new methodological tool that will help with the following:-Understanding how much dialogic practices matter to healthcare professionals.-Testing the proposition that RC and DP are interrelated and mutually reinforcing. This will allow connections to be made between different approaches to the problem of coordination, recognising its influence on teamwork quality and performance.-Other researchers to detect the presence and intensity of DPs in healthcare teams or in other contexts. This will allow for comparative studies and the possible generalisation of findings obtained in healthcare case studies.

Therefore, as intended in the main objective of study, a tool to identify and measure DP has been validated. This can be a helpful tool for healthcare organisations to utilise, along with alternative guidelines, beyond medical protocols to help them act whenever an unforeseen situation appears.

## Figures and Tables

**Figure 1 healthcare-11-02961-f001:**
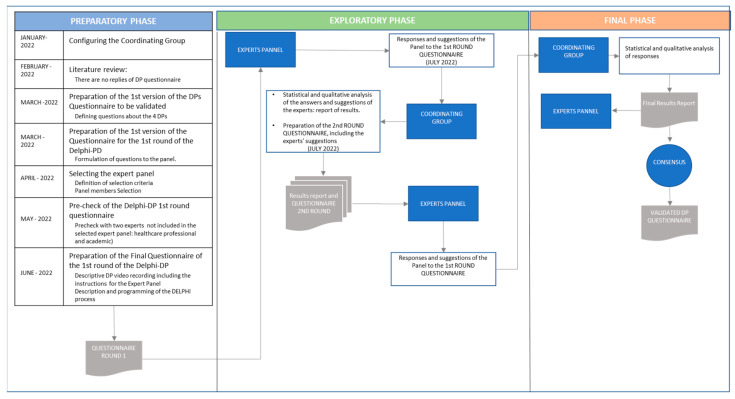
Stages in the Delphi method.

**Table 1 healthcare-11-02961-t001:** Panel of experts who participated in the study.

ID	Professional Profile	Years of Experience	Academic Qualifications
1	Emergency Physician	+25 years	Degree in Medicine
2	Deputy Director of Nursing	+25 years	Graduate in Nursing
3	General Nursing Supervisor	+25 years	Graduate in Nursing
4	Associate Professor of Nursing	+12 years	PhD in Nursing
5	ICU Nurse and Supervisor	+25 years	Graduate in Nursing
6	Medical Internist	+25 years	Degree in Medicine
7	Internal Medicine Nurse	+25 years	Graduate in Nursing
8	University Professor of Strategy and Leadership	+12 years	PhD in Business Organisation
9	University Professor of Preventive Medicine	+25 years	Doctor of Medicine and Surgery
10	Out-of-Hospital Emergency Physician	+25 years	Degree in Medicine

**Table 2 healthcare-11-02961-t002:** Criteria for measuring the degree of consensus taken from [47].

	Level of Consensus	
(Agreement) (A)	Neutral (N)	Disagreement (D)
If Mdn ≥ 4 and IQR ≤ 1.5		Mdn ≤ 3.5 and IQR ≤ 1.5
or	If Mdn ≥ 3.5 and IQR ≤ 2	or
If Mdn ≥ 4 and IQR ≤ 2 and *f* (4–5) ≥ 70%		Mdn ≤ 3.5 and IQR ≤ 2 and *f* (1–3) ≥ 70% ^1^

^1^ Mdn = Median; *f* = Frequency; IQR = Interquartile Range.

**Table 3 healthcare-11-02961-t003:** Round I Delphi Results.

						Quartile						
		Mdn	Mean	SD	CV	1	2	3	IQR	RIR	*f* (4–5)	CONS ^1^
DP1	DP Item	4.00	3.70	0.64	0.17	3.75	4.00	4.00	0.25	0.06	80%	A
	Scale Item	4.00	3.90	0.70	0.18	4.00	4.00	4.00	0.00	0.00	90%	A
DP2	DP Item	4.00	4.10	0.54	0.13	4.00	4.00	4.25	0.25	0.06	90%	A
	Scale Item	4.00	4.10	0.54	0.13	4.00	4.00	4.25	0.25	0.06	90%	A
DP3	DP Item	4.00	4.10	0.54	0.13	4.00	4.00	4.25	0.25	0.06	90%	A
	Scale Item	4.00	4.30	0.46	0.11	4.00	4.00	5.00	1.00	0.25	100%	A
DP4	DP Item	4.00	3.90	0.83	0.21	3.75	4.00	4.25	0.50	0.13	80%	A
	Scale Item	4.00	4.10	0.54	0.13	4.00	4.00	4.25	0.25	0.06	90%	A

^1^ Mdn = Median; Mean = Arithmetic Mean; SD = Standard Deviation; CV = Coefficient of Variation; IQR = Interquartile Range; RIR = Relative Interquartile Range; *f* (4–5) = Frequency of Assessment 4 and 5; CONS = Level of Consensus.

**Table 4 healthcare-11-02961-t004:** Experts’ qualitative feedback in the first round.

DP ^1^	Items	Feedback	Response Coordinating Group	Rewording of the Item
DP1	1.1 In a critical and emergency setting, have you witnessed or been involved in situations in which the patient’s evolution has led to differences of opinion on the treatment to be followed between different specialities/profiles? 1.2 In these cases, is a consensus reached on the treatment to be followed?	1.1. “I would replace different specialities-profiles with different professionals, as discrepancies can also arise between professionals in the same speciality”.	The suggestion in 1.1 is accepted.	1.1 In a critical and emergency setting, have you witnessed or been involved in situations in which the patient’s evolution has led to differences of opinion on the treatment and/or care to be followed between different professionals? 1.2 In these cases, is a consensus reached on the treatment and/or care to be followed?
		1.2. The expert asks the following question: “In which situations does the patient’s evolution give rise to a difference of opinion between professionals: in common pathologies, in rare pathologies, in simple diseases, in medical complications?”	We understand that the expert has asked to replace the original question with the previous question. In this case, we consider that the original question is about whether this dialogic practice has taken place but not the specific situations that trigger it. Nevertheless, we consider that this question is very interesting and could be added in a future study.	
		1.1. and 1.2. “Add treatment and/or care in both questions”.	The suggestion in 1.1. and 1.2. is accepted.	
DP2	In situations that do not respond as expected, do the following interveners (colleagues from your own speciality or colleagues from other specialities and/or other profiles) ask you for help in deciding how to deal with a clinical case?	“Replace in situations that do not respond as expected within unforeseen situations”	The suggestion is accepted.	In unforeseen situations, do the following interveners (colleagues from your own speciality, or colleagues from other specialities and/or other profiles) ask you for help or advice when deciding how to deal with a clinical case?
DP3	Do you usually remind, caution, or alert the following profiles about any important actions that by action or omission may compromise the quality of the intervention?	“I would replace the following profiles with other professionals in your environment of the same or different professional category. To obtain a more realistic answer, the two options should even be separated.	The terminological suggestion is accepted. However, it is not envisaged to separate the two options, since in the final survey, the different professional profiles will be indicated.	Do you usually remind, caution, or alert other professionals in your environment, of the same or different professional category, about any important action that by action or omission may compromise the quality of the intervention?
		The profiles should be added, as they are not detailed”.	Professional profiles will be indicated in the final survey.	
DP4	In emergency scenarios, have you witnessed or been involved in situations where it has been necessary to act outside of any of the established protocols?	“There should be a supplementary question for the specific case of the COVID-19 pandemic: In emergency scenarios, have you witnessed or been involved in situations where it has been necessary to act in the absence of established protocols for that specific circumstance?”	It does not consider including a specific question, as the final survey will differentiate between COVID-19 and non-COVID-19 contexts. However, the suggestion is considered in terms of the wording of the question, distinguishing between actions in the absence of protocols and outside the protocol.	In emergency scenarios, have you witnessed or been involved in situations in which it has been necessary to act without reference protocols or outside any of the established protocols?

^1^ DP1 = Epistemic contestation; DP2 = Joint sensemaking; DP3 = Cross-boundary intervention; DP4 = Protocol breaking.

**Table 5 healthcare-11-02961-t005:** Round II Delphi Results.

						Quartile						
		Mdn	Mean	SD	CV	1	2	3	IQR	RIR	*f* (4–5)	CONS ^1^
DP1	DP Item	4.50	4.50	0.50	0.11	4.00	4.50	5.00	1.00	0.22	100%	A
DP2	DP Item	4.00	4.30	0.64	0.15	4.00	4.00	5.00	1.00	0.25	90%	A
DP3	DP Item	4.50	4.50	0.50	0.11	4.00	4.50	5.00	1.00	0.22	100%	A
DP4	DP Item	5.00	4.60	0.49	0.11	4.00	5.00	5.00	1.00	0.20	100%	A

^1^ Mdn = Median; Mean = Arithmetic Mean; SD = Standard Deviation; CV = Coefficient of Variation; IQR = Interquartile Range; RIR = Relative Interquartile Range; *f* (4–5) = Frequency of Assessment 4 and 5; CONS = Level of Consensus.

**Table 6 healthcare-11-02961-t006:** Comparison of results in rounds I and II.

					Quartile					
DP	Rounds	Mdn	Mean	SD	1	2	3	IQR	*f* (4–5)	CONS ^1^
DP1	R1	4.00	3.73	0.67	3.75	4.00	4.00	0.25	80%	A
	R2	4.50	4.50	0.53	4.00	4.50	5.00	1.00	100%	A
DP2	R1	4.00	4.09	0.57	3.00	4.00	4.00	1.00	90%	A
	R2	4.00	4.27	0.67	4.00	4.00	5.00	1.00	90%	A
DP3	R1	4.00	4.27	0.48	4.00	4.00	4.00	0.00	100%	A
	R2	4.50	4.50	0.53	4.00	4.50	5.00	1.00	100%	A
DP4	R1	4.00	4.09	0.57	3.00	4.00	4.00	1.00	90%	A
	R2	5.00	4.64	0.52	4.00	5.00	5.00	1.00	100%	A

^1^ Mdn = Median; Mean = Arithmetic Mean; SD = Standard Deviation; IQR = Interquartile Range; *f* (4–5) = Frequency of Assessment 4 and 5; CONS = Consensus Level.

**Table 7 healthcare-11-02961-t007:** Variation of RIR and CV between rounds.

	DP1	DP2	DP3	DP4
Variation of CV ^1^	0.06	−0.02	0.02	0.11
Variation of RIR ^2^	−0.16	−0.19	−0.16	−0.08

^1^ $Variation\;of\;CV\;={\left(\frac{SD}{Mean}\right)}_{Round1}-{\left(\frac{SD}{Mean}\right)}_{Round2}$ ^2^ $Variation\;of\;RIR={\left(\frac{Q3-Q1}{Q2}\right)}_{Round1}-{\left(\frac{Q3-Q1}{Q2}\right)}_{Round2}$.

## Data Availability

The data that support the findings of this research are available on request from the corresponding author [M.-d.-R.P.-O.].

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
