# Peer review of "How Much Dialogic Coordination Practices Matter to Healthcare Professionals—A Delphi Approach towards a Tool for Identification and Measurement"

_healthcare, 2023, doi:10.3390/healthcare11222961_

Round 1

Reviewer 1 Report

Comments and Suggestions for Authors

Title: How much Dialogic Coordination Practices matter to healthcare professionals.

Abstract: Abstract: The study of coordination practices in health policy and, especially in emergency situations is a central aspect. The COVID-19 pandemic has highlighted the need for further research on such practices, especially those that are deployed in complex and unforeseen situations to respond quickly while minimising errors. These dialogic coordination practices (DP) have been identified through case studies but have not been validated. The purpose of this study is to develop and validate a DP questionnaire for healthcare teams. The development of the items was based on a literature review and the content validation was carried out by means of a Delphi study. A total of 10 experts assessed the clarity and appropriateness of the items and their corresponding measurement scales. Consensus was reached after the second round, with a percentage of agreement greater than or equal to 90% for all items. The validation of these items constitutes a methodological innovation that responds to the call in the literature to open up new avenues for comparative studies, the possibility of generalising the findings obtained and the possibility of bringing together different approaches to the problem of coordination, key in health policy.

 Dear Authors ,

Good day to you. Overall, the paper focuses on and discusses about the The study aimed to create and validate a questionnaire for identifying four types of teamwork challenges in hospitals: epistemic contestation, joint sensemaking, cross-boundary intervention, and protocol breaking. Validation before questionnaire launch is crucial for data quality, reliability, and relevance to the surveyed population. The Delphi method, a valuable research approach, is especially useful in areas with limited or emerging knowledge, like social and health sciences. It's commonly used to validate various questionnaire types. While there aren't strict quality criteria, key parameters include expert selection, anonymity, iteration, feedback control, and statistical consensus stability. Your work is commendable, and these observations have been made to improve the quality of the article further.

·       The abstract needs improvements, particularly by discussing the research's outcomes.

·       It is essential to clarify how the sample size was determined with clear justification.

·       To enhance clarity, the methodology section would greatly benefit from including a flowchart.

·       To enhance the result discussion, exploring advanced data analysis techniques is advisable.

·       The conclusion should be refined to convey a more professional tone and enhance clarity.

Best Wishes

Author Response

Dear reviewer,

We appreciate the effort of the review and your observations, and we communicate that we have taken them into account in the new version of article. Your feedback has allowed us to improve our manuscript. In the following paragraphs you can see the answers to each of your recommendations and thank you very much for your words of support, greetings.

C1:  The abstract needs improvements, particularly by discussing the research's outcomes

According to the requests of the reviewer, the abstract has been enhanced by including a more comprehensive overview of the research findings. New information is in yellow in text.

C2:  It is essential to clarify how the sample size was determined with clear justification.

In the description of the steps followed during the preliminary phase of the Delphi, a paragraph has been included justifying the choice of the Delphi expert panel size. New information is in yellow in text.

C3:  To enhance clarity, the methodology section would greatly benefit from including a flowchart.

The flowchart is presented in Figure 1, in the Materials and Methods section. Do you believe this figure should appear at the beginning of this section?

C4: To enhance the result discussion, exploring advanced data analysis techniques is advisable. 

We appreciate your comment on the analysis techniques used in our study.

We would like to highlight that we have adhered to the standard practices of Delphi methodology in questionnaire validation. The high level of agreement among experts in our study suggests that more advanced analysis techniques may not provide substantial additional value. While there is no single methodology or approach for data analysis in Delphi studies in the scientific literature, we believe our current approach aligns with standard practices in the field and has been compared to similar studies. Nevertheless, we value your suggestions for improving the quality of our research.

Deeper statistical analysis will be conducted when the full questionnaire, incorporating the CR and PD questions, is launched. But this is future research in this project and we have included this information in the conclusions.

C5: The conclusion should be refined to convey a more professional tone and enhance clarity.

According to the requests of the reviewer, the conclusion has been rephrased, and we hope to have enhanced the clarity and rigor of the text.

Reviewer 2 Report

Comments and Suggestions for Authors

The article "How much Dialogic Coordination Practices matter to healthcare professionals" appears to address an important topic in the context of healthcare, particularly in light of the COVID-19 pandemic. However, there are several areas where the article could be improved:

Title:

The title could be more specific and informative. It should clearly convey the study's focus and purpose, allowing readers to understand what to expect from the article.

Clarity and Structure:

The abstract should provide a clear and concise summary of the study's objectives, methodology, and key findings. This abstract is quite vague and lacks specific details about the research.

Research Gap and Rationale:

The article mentions the need for research on dialogic coordination practices, especially in emergency situations, but it could benefit from a more detailed discussion of the existing literature and the specific research gap this study aims to address. What do we already know, and what is this study adding to our understanding?

Research Objectives:

The article should clearly state the research objectives and research questions it seeks to answer. What is the purpose of developing and validating the DP questionnaire for healthcare teams? What are the anticipated outcomes? you should write this at the end of the introduction.

Methodology:

clear and comprehensive 

Sample Size and Selection:

clear and comprehensive 

Results and Implications:

The article should provide a glimpse of the study's findings or the significance of reaching a consensus on the items. Without this, the reader is left wondering about the implications and practical applications of the research.

In summary, this work needs further development to provide a clearer picture of the research's objectives, methods, and findings. Adding more specific information and context would make it more informative and engaging for readers in the healthcare and research communities.

Author Response

We appreciate the effort of the review and your observations, and we communicate that we have taken them into account in the new version of article. Your feedback has allowed us to improve our manuscript. In the following paragraphs you can see the answers to each of your recommendations and thank you very much for your words of support, greetings.

C1:  The title could be more specific and informative. It should clearly convey the study's focus and purpose, allowing readers to understand what to expect from the article

The title has been modified, and we hope it better reflects the study's focus and purpose.

C2:  The abstract should provide a clear and concise summary of the study's objectives, methodology, and key findings. This abstract is quite vague and lacks specific details about the research.

According to the requests of the reviewer, the abstract has been enhanced by including a more comprehensive overview of the research findings. New information is in yellow in text.

C3:  The article mentions the need for research on dialogic coordination practices, especially in emergency situations, but it could benefit from a more detailed discussion of the existing literature and the specific research gap this study aims to address. What do we already know, and what is this study adding to our understanding?

In the introduction, several paragraphs (highlighted in yellow) have been added to clarify these aspects.

C4: The article should clearly state the research objectives and research questions it seeks to answer. What is the purpose of developing and validating the DP questionnaire for healthcare teams? What are the anticipated outcomes? you should write this at the end of the introduction.

At the end of the introduction, a paragraph has been added (in yellow), describing the usefulness and purpose of this research, as well as the expected outcomes.

C5: Results and Implications: The article should provide a glimpse of the study's findings or the significance of reaching a consensus on the items. Without this, the reader is left wondering about the implications and practical applications of the research.

At the end of the article in Discussion and Conclusion, a glimpse of the study’s findings has been included (in yellow):

C5: In summary, this work needs further development to provide a clearer picture of the research's objectives, methods, and findings. Adding more specific information and context would make it more informative and engaging for readers in the healthcare and research communities.

We appreciate your comment on the analysis techniques used in our study.

We have changed some parts/words to provide a clearer picture of objectives, methods, and findings (in yellow).

We would like to highlight that we have adhered to the standard practices of Delphi methodology in questionnaire validation. The high level of agreement among experts in our study suggests that more advanced analysis techniques may not provide substantial additional value. While there is no single methodology or approach for data analysis in Delphi studies in the scientific literature, we believe our current approach aligns with standard practices in the field and has been compared to similar studies. Nevertheless, we value your suggestions for improving the quality of our research.

Deeper statistical analysis will be conducted when the full questionnaire, incorporating the CR and PD questions, is launched. We have included this information in text.

C5: The conclusion should be refined to convey a more professional tone and enhance clarity.

According to the requests of the reviewer, the conclusion has been rephrased, and we hope to have enhanced the clarity and rigor of the text.